# Generation of Rapid and High-Quality Serum by Recombinant Prothrombin Activator Ecarin (RAPClot™)

**DOI:** 10.3390/biom14060645

**Published:** 2024-05-30

**Authors:** Kong-Nan Zhao, Goce Dimeski, Paul Masci, Lambro Johnson, Jingjing Wang, John de Jersey, Michael Grant, Martin F. Lavin

**Affiliations:** 1Australian Institute of Biotechnology and Nanotechnology, The University of Queensland, Brisbane, QLD 4072, Australialambro.johnson@uq.edu.au (L.J.); jingjing.wang2@uq.net.au (J.W.); 2Chemical Pathology, Princess Alexandra Hospital, Woolloongabba, Brisbane, QLD 4102, Australia; goce.dimeski@health.qld.gov.au; 3School of Chemistry and Molecular Biosciences, The University of Queensland, Brisbane, QLD 4072, Australia; j.dejersey@uq.edu.au; 4School of Medicine, University of Queensland, Brisbane, QLD 4072, Australia; 5Q-Sera Pty Ltd., Level 9, 31 Queen St, Melbourne, VIC 3000, Australia; michael.grant@q-sera.com; 6Centre for Clinical Research, The University of Queensland, Brisbane, QLD 4029, Australia

**Keywords:** prothrombin activator, RAPClot™ prototype tube, γ-radiation, enzymatic activity, blood clotting, serum, biochemical analytes

## Abstract

We recently reported the potential application of recombinant prothrombin activator ecarin (RAPClot™) in blood diagnostics. In a new study, we describe RAPClot™ as an additive to develop a novel blood collection prototype tube that produces the highest quality serum for accurate biochemical analyte determination. The drying process of the RAPClot™ tube generated minimal effect on the enzymatic activity of the prothrombin activator. According to the bioassays of thrombin activity and plasma clotting, γ-radiation (>25 kGy) resulted in a 30–40% loss of the enzymatic activity of the RAPClot™ tubes. However, a visual blood clotting assay revealed that the γ-radiation-sterilized RAPClot™ tubes showed a high capacity for clotting high-dose heparinized blood (8 U/mL) within 5 min. This was confirmed using Thrombelastography (TEG), indicating full clotting efficiency under anticoagulant conditions. The storage of the RAPClot™ tubes at room temperature (RT) for greater than 12 months resulted in the retention of efficient and effective clotting activity for heparinized blood in 342 s. Furthermore, the enzymatic activity of the RAPClot™ tubes sterilized with an electron-beam (EB) was significantly greater than that with γ-radiation. The EB-sterilized RAPClot™ tubes stored at RT for 251 days retained over 70% enzyme activity and clotted the heparinized blood in 340 s after 682 days. Preliminary clinical studies revealed in the two trials that 5 common analytes (K, Glu, lactate dehydrogenase (LD), Fe, and Phos) or 33 analytes determined in the second study in the γ-sterilized RAPClot™ tubes were similar to those in commercial tubes. In conclusion, the findings indicate that the novel RAPClot™ blood collection prototype tube has a significant advantage over current serum or lithium heparin plasma tubes for routine use in measuring biochemical analytes, confirming a promising application of RAPClot™ in clinical medicine.

## 1. Introduction

It is well documented that laboratory blood tests impact at least 70% of the patient care decisions starting from diagnosis, treatment, and management to discharge [1,2]. Over the past two decades, there have been additional changes in the tubes used for blood collection for laboratory tests to improve the accuracy of results. This includes the introduction of thrombin as a clotting agent [3,4], a new cell plasma separator device to decrease cell remnants in the plasma of lithium heparin plasma [4] and new anticoagulant combinations to prevent glucose consumption by cells [5].

Currently, there are a number of tubes that produce either serum or plasma for analyte testing. The most frequently used fluid–serum or plasma tubes are as follows: (a) serum tubes with or without gel separator and either with silica or thrombin as the procoagulant which require time to clot and do not always provide fully clotted samples in patients on anticoagulants and (b) lithium heparin tubes suitable for immediate centrifugation to improve the turn-around-times of results (TAT). Both of these types of tubes have well-documented problems in producing desirable high-quality samples [6,7,8,9]. The vast majority of results in chemical pathology and serology testing are obtained from serum. There are only a few analytes that require plasma samples [10,11].

Although rapid analysis is achieved with plasma, the presence of clotting factors and higher concentrations of remnant cellular material post-separation of the cells from plasma can alter the integrity and stability of the sample upon short or prolonged storage, compromising the accuracy of many critical analytes [12]. It has also been reported that the presence of anticoagulants in plasma collection tubes can introduce interfering factors, such as enzyme inhibitors, fibrinogen and cations [13,14], and fibrins, such as with the Beckman troponin assay [15]. The alternative, serum, represents a cleaner, higher-quality sample type when blood is fully clotted before centrifugation. However, the standard commercially available serum tubes, primarily using silica particles for the coagulation of blood, require long clotting times (30 min) for healthy individuals and longer times (over 60 min) for patients on anticoagulants, whose samples may only achieve partial or no clotting at all (Dimeski PhD Thesis) [3,7,12,16,17]. The use of rapid serum tubes (RST) containing bovine thrombin is a well-established technology for serum preparation [3,4]. The drawback of these tubes is their high costs and inability to clot blood from many anticoagulated patients [3,4].

To overcome the above-described issues, we have employed snake venom prothrombin activators (PAs) for the rapid preparation of consistently high-quality serum in both normal individuals and in blood from anticoagulated patients [18,19]. PAs utilize prothrombin in the blood samples to generate rapid and sustained levels of human thrombin, and more thrombin is produced than the amount used in commercially available thrombin tubes [18,19]. We have established this using PAs purified from the venoms of *Oxyuranus scutellatus* (OsPA) and *Pseudonaja textilis* (PtPA) added to blood collection tubes to efficiently coagulate blood from normal and several anticoagulated samples [18,19]. While these venoms contain relatively large amounts of PAs [20,21], it was apparent that their use in commercial tubes was not optimal due to supply considerations associated with procuring venom from snakes. Although venom-sourced proteins can be produced to a high purity, it was desirable to develop a recombinant form of snake venom PA for more consistent quality generation and in sufficient quantity, which could be manufactured in a controlled environment of high purity in any quantity with quality assured. The PA ecarin from the saw-scaled viper (*Echis carinatus*) was selected because unlike OsPA and PtPA, it is synthesized as a single polypeptide chain and is subject to a lesser amount of post-translational modification [22,23]. Ecarin has a very different structure than OsPA and PtPA which consist of active factor Xa- and factor Va-like proteins in a stable complex, homologous with the human prothrombinase complex [23,24]. Ecarin is a metalloproteinase and like all snake venom PAs is minimally or not affected by any regulatory components of the mammalian coagulation–fibrinolysis system including activated protein C and antithrombin III (ATIII) [23,25], making it ideal for rapid clotting of blood including anticoagulated blood, producing high-quality serum for analysis.

Recently, we reported that a codon-optimized form of ecarin was successfully cloned for expression in mammalian cells at high yield [23]. It was demonstrated that the recombinant ecarin enzyme could efficiently clot normal blood and blood spiked with high concentrations of anticoagulants including heparin and had great potential as an additive to blood collection tubes to produce high-quality serum for analyte testing in diagnostic medicine [23]. Here, we describe an experimental approach to develop a novel type of RAPClot™ prototype tube for future application in analyte determination in diagnosis.

## 2. Materials and Methods

### 2.1. RAPClot™ Prototype Tubes and Commercial Blood Collection Tubes

Firstly, Greiner-Bio-One (GBO,) white-top no-additive plain blood collection tubes (GBO Vacuette^®^ Catalogue No 4566001, Kremsmünster, Austria) were used to prepare RAPClot™ prototype tubes for laboratory experiments. Twenty microliter of 0.24% surfactant (Dow Corning silicone hydrophilic surfactant (Catalogue No SH3771 H, Midland, MI, USA)) was used to coat the bottom of the tubes and then they were dried. Becton Dickinson (BD) red-top no-additive plain blood collection tubes (BD Vacutainer^®^ Catalogue No 366406, Franklin Lakes, NJ, USA) and Greiner BCA Fast Clot tubes (GBO Vacutainer^®^ Catalogue No 456313, Kremsmünster, Austria) were also used in some experiments where indicated.

RAPClot™ concentrate was added to 20 µL of the patented protective formulation consisting of a colloid, Gelofusine^TM^ (succinylated gelatin 4%-GF, Catalogue No 210317641, Sydney, Australia) with a stabilizing sugar which was added to the surfactant-coated tubes and dried using an air–nitrogen dryer (Brisbane, Australia). Dried prototype tubes treated with γ-radiation at ~25–27.8 kGy were labelled as RAP+Ir, with untreated tubes labelled as RAP-Ir (Appendix A). In tubes designated “wet”, an aliquot of RAPClot™ concentrate was added to the blood collection tube without drying prior to adding the blood sample.

Three types of BD commercially available blood collection tubes that included BD standard serum separator tube (SST, Catalogue No 367974) with silica as clot activator, BD rapid serum tubes containing bovine thrombin (RST Catalogue No 368771), and BD Vacutainer^®^ PST™ tube (PST Catalogue No 367962) (Brisbane, Australia) were used in clinical trials.

### 2.2. Study Design

To use RAPClot™ as an additive to develop a novel type of blood collection tube for clinical diagnosis, we designed and carried out different types of experiments in the laboratory. (A). Formulation development experiments: Different colloids such as lactulose, dextran, Polyvinylpyrrolidone, voluven, and sorbitol were initially tested for developing an optimal RAPClot™ formulation which is particularly helpful for clotting anticoagulant-blood, especially heparin-blood, to produce high-quality serum (Appendix A). Finally, six RAPClot™ formulations were designed for developing RAPClot™ prototype tubes for further experiments (Appendix A). (B). γ-radiation and Electron-beam (E-beam) sterilization experiments: RAPClot™ prototype tubes prepared in different formulations were dried in nitrogen air at room temperature. The RAPClot™ prototype tubes were then divided into two groups, one group of the tubes was sterilized by either γ-radiation (Appendix A) or *E-beam* with a typical dose range of 25–30 kGy, and the other group was used as control without γ-radiation and E-beam sterilization. Both types of the prototype tubes were assayed for the activity of RAPClot™ by S2238 assay and blood clotting assay. (C). Shelf-life experiments of RAPClot™ prototype tubes: Currently, most blood collection tubes on the market have at least a 12-month shelf-life (Ref). Thus, the prototype tubes whether they were dry-only or sterilized by γ-radiation/E-beam were also divided into two groups, which were, respectively, stored under two temperature conditions—room temperature (RT) and higher temperatures (50 °C) that could cause reductions in draw volume up to two years. (D). Small clinical trials: Trial 1 was designed to recruit five volunteers for assessing the capacity of the γ-sterilized RAPClot™ prototype tubes in clotting both fresh and heparinized blood and its effects on the determination of five important analytes. Trial 2 was designed to determine all 33 analytes in sera produced by RAPClot™ tubes from the five volunteers, compared to those produced by three commercial blood collection tubes.

### 2.3. Clotting of Whole Blood Samples in Blood Collection Tubes

Either fresh whole blood was added directly to tubes or for recalcified citrated whole blood, 50 µL of 1 M CaCl_2_ was added followed by 3.95 mL of citrated whole blood (total sample volume 4 mL). The tubes were recapped immediately after the timer was started and gently tilted every 15 s for 5–6 times to monitor clotting. Clotting start times were estimated visually and recorded when the clotting was first observed and when a firm clot formed as defined by the clot staying in place upon the inversion of the tube as previously reported [18,19]. For experiments using anticoagulated blood, fresh whole or recalcified citrated whole blood was spiked with commercial sodium heparin solution (DBL^TM^ heparin sodium injection BP, C84593, Pfizer, New York, NY, USA) and dosed as above.

### 2.4. Plasma Clotting Assay

The recalcified citrated plasma clotting assay was performed using a Hyland-Clotek instrument as described previously [19].

### 2.5. Thrombelastography (TEG) of Recalcified Citrated Whole Blood

The TEG^®^ Haemostasis Analyser 5000 series (Haemscope Corporation, Niles, IL, USA) was used as per manufacturer recommendations and as described elsewhere [26]. The TEG assay captures four important parameters (R time, K time, α-angle, and MA value). The R-value represents the time until the first evidence of a clotting; the K value is the time from the end of R until the clot reaches 20 mm, representing the speed of clot formation; the α-angle is the tangent of the curve made as the K is reached, and MA is a reflection of clot strength [27]. The details of the TEG assay were the same as described previously [18,19].

### 2.6. S2238 Chromogenic Bioassay for RAPClot™ Prototype Tube

RAPClot™ was assayed in a single reaction mixture containing prothrombin and the thrombin-specific substrate S2238 (Cat No 00082032439, Werfen, Barcelona, Spain). The coupled reactions, prothrombin to thrombin and S2238 to pNA, monitored at 410 nm, result in the non-linear progress curves of absorbance vs. time. These were analyzed by fitting second-order polynomials to give the RAPClot™ activity in mUs defined as nmol thrombin/min, and the assay was also carried out to monitor the recovery of RAPClot™ in RAPClot™ prototype tubes [28].

### 2.7. Statistical Analysis

The Excel-2403 (Formula-Statistical program) software was used for all analyses with Student’s two-tailed *t*-test and one-way analysis of variance being employed. The patients and sample numbers together with numerical values including the mean ± standard deviation are included. *p* values < 0.05 or *p* < 0.01 were used to present the significance levels.

### 2.8. Human Research Ethics

The study was conducted in accordance with the Declaration of Helsinki. Human research ethics approval for this study involving blood collection from volunteers and patients was obtained from Metro South Human Research Ethics Committee and The University of Queensland Human Ethics Committee: HREC Reference number: HREC/08/QPAH/005. Most recent date of approval on 21 March 2017. The supply of human blood for research with ethics approval was obtained from the Australian Red Cross Service (ARCBS), Brisbane, Australia.

## 3. Results

### 3.1. Establishment of Stabilizing Formulation to Develop RAPClot™ Prototype Tube

To be commercially viable as a procoagulant in blood collection tubes, RAPClot™ must retain its activity over standard industry manufacturing and storage stability conditions [29,30]. We initially observed that wet RAPClot™, dissolved in Hepes or Gelofusine (GF) patented buffer and added to GBO or BD plain tubes (Figure 1A), coated with either of the two silicone surfactants DC3771 or L7-9245 (Dow Corning, Midland, MI, USA) had similar blood clotting times that varied from 92 to 98 s (Figure 1A). By comparison, control tubes without RAPClot™ took longer than 30 min to clot (Figure 1A). Whole blood clotting was shown to depend on the concentration of RAPClot™ without any influence by the surfactant (Appendix A). To establish a stabilizing formulation for developing RAPClot™ prototype tubes coated with or without the surfactant DC3771, the RAPClot™ was dried in the tubes in both the GF and Hepes buffers under standard laboratory conditions (Figure 1B). The clotting activity of the air-dried RAPClot™ in the GF buffer was comparable in time to that observed with the wet RAPClot™ in both the GBO and BD tubes and the surfactant had no effect on the clotting (Figure 1B). However, when the Hepes buffer was used instead of the GF buffer, the clotting activity of the air-dried form was markedly reduced (Figure 1B). These findings suggested that the RAPClot™ tested in the GF buffer whether it was wet or air-dried in different tube types has great potential for development as a rapid serum prototype tube. We also tested bovine serum Albumin (BSA, Sigma-Aldrich, St Louis, MO, USA), Polyvinylpyrrolidone (PVP, Merck, Darmstadt, Germany), Voluven^TM^ (a starch-based plasma volume expander) (Fresenius Kabi Ltd., Sydney, Australia), dextran, and several stabilizing sugars (sorbitol sucrose, trehalose and mannitol) that might play an important role in stabilizing RAPClot™ (Appendix A). The results show that the addition of different compounds had some effects on activity with a combination of Gelofusine and 10% stabilizing sugar having the most significant effect on stabilizing activity. We employed the GF stabilizing formulation with RAPClot™ and dried the tubes by nitrogen and vacuum-desiccator air-dry to determine the drying process impact on the enzymic activity of RAPClot™. The use of the S2238 chromogenic bioassay showed that the process of drying had no effect on the RAPClot™ activity compared with the wet RAPClot™ solution (Figure 1C). In the next experiment, RAPClot™ at 0.15 mU/tube was added with the identified stabilizing formulation to the tubes that were dried by vacuum-desiccator air-drying, with results showing no loss of the blood clotting activity consistent with the results from the S2238 assay (Figure 1D). The results again confirmed that our patented GF formulation (Patent No, WO2016061611A1) was suitable for developing RAPClot™ blood collection prototype tubes for the generation of high-quality serum. We defined the quality of the serum based on the visual lack of cellular material and red blood cell hang-up, lack of fibrin strands as well as fibrinogen (Appendix A). This shows the quality of the serum produced in the RAPClot™ tubes compared to four other commercial tubes. In addition, the determination of five key analytes was comparable to those measured in a commercial blood collection tube in a small clinical trial. This also agrees with the data obtained in another clinical trial showing that the serum quality produced by the RAPClot™ prototype tube does not interfere with the determination of 33 different biochemical analytes (Appendix A).

### 3.2. γ-Radiation Sterilization of the RAPClot™ Prototype Tube

The RAPClot™ prototype tubes (0.33 mU/tube) were prepared in four formulations (Appendix A) and treated with or without γ-radiation (25.34 kGy) as the sterilizing agent to determine the effect on the enzymatic activity (Figure 2A). We observed that the RAPClot™ in formulation A (S-A, lead formulation) did not lose activity after drying; however, the activity was reduced by 8–15% in the other three formulations (S-B, S-C and S-D) compared with the wet solution acting as the control (Figure 2A). Subsequently, the γ-radiation treatment reduced the RAPClot™ activity by 26% in the tubes dried with formulation A and 32–38% in the other three formulations (Figure 2A), providing further evidence that our preferred formulation (A) was the most suitable for developing RAPClot™ prototype tubes. Next, we prepared RAPClot™ 0.4 mUnit/4 mL blood prototype tubes with formulation A to which we added an increasing concentration of BSA followed by sterilization with γ-radiation (27.8 kGy) and compared these to the RAPClot™ in the Hepes buffer with the addition of lactulose and BSA (Appendix A; Figure 2B). Using a plasma clotting assay (Appendix A), the activity of the RAPClot™ tubes without sterilization decreased from 97% to 76% (21% decrease) with the increasing concentration of BSA while that for the tubes treated with γ-radiation decrease ranged from 69% to 66% (Figure 2B). In comparison with the S2238 enzymatic assay, the untreated RAPClot™ tubes’ decrease ranged from 79% to 66% with increasing concentrations of BSA while those treated with γ-radiation had a decrease range of 55% to 50% (Figure 2B). Thus, the decrease with added BSA was of the same order for both clotting and enzyme activity assays in each case, revealing that the two assays are correlated (r = 0.9404**, Figure 2C). A related set of data comparing clotting times to the S2238 activity with increasing BSA protein in the Hepes buffer also showed a similar relationship (Figure 2B). The effect of the heparin anticoagulant on the RAPClot™ tubes using recalcified citrated whole blood on day 15 post-radiation showed clotting was achieved in <2.5 min without heparin and in the presence of 8 U/mL heparin clotting was achieved in <5 min (Figure 2D). All the RAPClot™ tubes generated clearly high-quality serum without latent clotting occurring at 24 h after centrifugation (Figure 2E). These results suggest that the RAPClot™ prototype tubes using formulation A (GF + 10% *w*s*v* stabilizing sugar) retained the highest blood clotting and S2238 activity after the γ-radiation sterilization at a dose of 27.8 kGy.

### 3.3. TEG Assay for the γ-Radiation-Sterilized RAPClot™ Blood Prototype Tubes

Based on the results obtained from the studies of γ-radiation-sterilized RAPClot™ prototype tubes in six formulations (Figure 2), the patented formation was used with the stabilizing sugar to prepare a new set of RAPClot™ prototype tubes containing two relatively lower doses of RAPClot™ tubes with 0.15 mU/tube (Figure 3) and 0.3 mU/tube (Appendix A). The newly produced RAPClot™ prototype tubes with or without γ-radiation at 25.7 kGy were stored at RT for testing blood clotting stabilities by the TEG assay at three time points over a two-month period (64 days) (Figure 3 and Appendix A). The TEG assay generates four whole blood clotting parameters: R time, K time, angle-α, and MA values (Figure 3A–D and Appendix A). As expected, the γ-radiation-treated prototype tubes showed slightly longer R times compared with the RAPClot™ wet control and the prototype tubes at 0.15 mU/tube without γ-radiation (Figure 3A). The R times were, however, similar with the heparinized blood at the three time points up to 64 days, indicating that the γ-irradiated tubes could retain the higher enzymatic activity in clotting the non-heparinized blood although they clotted the heparinized blood (4 U/mL), except in one sample where the result was 260 s at day 9 (Figure 3A) which was still within the reported clotting range of 263–496 s [31]. In a similar experiment using tubes as above with a dose of 0.3 mU/tube RAPClot™, the TEG R times improved with (4 U/mL) heparinized blood for up to 64 days (Appendix A). The γ-radiation-sterilized RAPClot™ prototype tubes containing 0.15 mU/tube generated K times for clotting the non-heparinized blood at the three time points similar to the RAPClot™ wet control and the non-irradiated prototype tubes (Figure 3B). Both RAPClot™ wet control and non-γ-radiation-sterilized (Dry-IR) prototype tubes had similar K times for clotting the heparinized blood over the 64 day period with the range of 55–160 s (Figure 3B). However, the γ-radiation-sterilized (Dry+IR) RAPClot™ prototype tubes produced a larger variation with the K times for clotting heparinized blood over the time course, with a time of 150 s at day 30 (Figure 3B), but showed an improved performance in the 0.3 mU-containing tubes (Appendix A). Both angle-α and MA values obtained from the TEG assay for the RAPClot™ prototype tubes are shown in Figure 3C,D. The Dry+IR RAPClot™ prototype tubes clotted heparinized blood producing a greater variation in angle-α and MA values, the lowest angle-α of 26.5 (<30) and MA value of 14.8 being on day 9 (Figure 3C,D), which were unacceptable for the propagation phase of coagulation and overall stability of the clot. However, the Dry+IR RAPClot™ prototype tubes containing 0.3 mU RAPClot™/tube clotted the heparinized blood at 4 U/mL with a significant increase in angle-α (46.5) and MA value (40.1) on day 9 (Appendix A), suggesting that a relatively higher dose of the RAPClot™ (>0.3 mU/tube) is required for the consistent clotting of heparinized blood to produce a solid blood clot with larger angle-α and higher mA values. Both the wet and Dry-IR RAPClot™ tubes produced satisfactory angle-α and MA values in clotting both non-heparinized and heparinized blood samples (Figure 3C,D and Appendix A).

The TEG demonstrated the maximum rate of thrombus generation (MRTG) values in mm/min for the individual RAPClot™ prototype tubes over the two-month period (Figure 3E and Appendix A). The results showed that the 0.15 mU/tube wet, Dry-IR, and Dry+IR tubes produced similar MRTG values in clotting non-heparin blood, although the MRTG values (16.72–18.84) at day 64 were significantly lower than those (25.11–25.62) at day 9 and (24.14–25.95) day 30 (Figure 3E and Appendix A). Correspondingly, all the tubes had very short TMRTG with a range of 2.33–2.83 min except the wet tube at day 30 which had only 1.83 min (Figure 3E,F; Appendix A). Furthermore, the results showed that all the wet and Dry-IR, especially RAP+IR, RAPClot™ tubes produced significantly lower MRTG values in clotting heparinized blood and the MRTG values for the Dry+IR tubes were only 3.55–8.77 at day 64 (Figure 3F, Appendix A). In addition, all the wet, RAP-IR, and RAP+IR sterilized RAPClot™ tubes had a high total thrombus generation (TTG) of 745.2–927.3 mm in clotting non-heparin blood but had a large variation in TTG (170.8–785.5 in clotting the heparinized blood, with the RAP+IR RAPClot™ tubes having the lowest TTG (Figure 3G and Appendix A). All the results revealed that heparin inhibited the MRTG, leading to prolonging the TMRTG and producing low TTG, suggesting that a relatively higher dose of RAPClot™ is required for developing the dry+γ-radiation RAPClot™ prototype tubes able to clot blood containing high concentrations of heparin.

### 3.4. Shelf Life (Blood Clotting Stability) of the Sterilized RAPClot™ Prototype Tube

Next, we investigated the potential shelf life of the sterilized RAPClot™ prototype tube. We prepared RAPClot™ -unirradiated prototype tubes (0.4 mU/tube) stored at room temperature (RT) for periods of up to 486 days to investigate their blood clotting activities (Figure 4A). The RAPClot™ tubes clotted efficiently recalcified citrated whole blood with or without heparin (8 U/mL) over the time course (Figure 4A). This was most apparent for the 0.4 mU/tube RAPClot™ tubes which clotted heparinized blood (8 U/mL) significantly faster than 300 s at day 486 (Figure 4A). Further experimentation showed that the RAPClot™ tubes containing 0.4 mU/tube still efficiently clotted both heparinized and non-heparinized blood after 3 years of storage (Appendix A). Two further batches of RAPClot™ prototype tubes (no radiation) were prepared and stored at both RT and 50 °C for 147 days (Figure 4B, left panel) and 286 days (Figure 4B, right panel), respectively. The 50 °C stored RAPClot™ tubes for 147 days (Figure 4B, left panel) and 286 days (Figure 4B, right panel) had very high activity in clotting heparinized blood (8 U/mL), with the clotting times of 194.5 and 133 s, respectively, comparable to the RT-stored tubes (Figure 4B). The results revealed that RAPClot™ appears to be stable at high temperatures and retains its activity in clotting high-dose heparinized blood. Furthermore, it demonstrated that the RAPClot™ prototype tubes containing a lower dose (0.26 mU/tube), which were sterilized with γ-radiation (27.8 kGy) and stored at RT, still exhibited high blood clotting capability up to day 373 (Figure 4C). After 12 months at RT, these tubes clotted heparinized blood (8 U/mL) in 342.5 s, only slightly longer than 320.5 s for non-irradiated tubes and 327 s for γ-radiation-sterilized tubes in clotting non-heparinized blood (Figure 4C). These results demonstrated that γ-radiation is useful for sterilizing RAPClot™ prototype tubes that maintain high blood clotting activity for greater than 12 months of storage.

### 3.5. Stability of RAPClot™ Prototype Tube Sterilized by Electron-Beam (E-Beam)

The effect of electron-beam (E-beam) radiation at a dose of 25 kGy was investigated on the enzymatic activity of the RAPClot™ prototype tubes, compared with that of γ-radiation (Figure 5). The S2238 assay showed a recovery of >90% enzymatic activity with the E-beam treatment, significantly higher than that of 59.5% with the γ-radiation treatment (Figure 5A). The next experiment was to compare the effects of E-beam with that of γ-radiation on the blood clotting activity of the RAPClot™ prototype tubes (Figure 5B). Exposure to E-beam radiation caused approximately a 10% loss of blood clotting activity, compared with a 40% loss for γ-radiation treatment (Figure 5B), consistent with the S2238 assay results (Figure 5A). The enzyme and blood clotting activities of the RAPClot™ prototype tubes were determined to contain a higher dose (0.9 mU/tube) with E-beam sterilization post storage at both RT and 50 °C conditions for 251 days, compared with those of the wet RAPClot™ prototype tubes stored at 4 °C and the dry-only RAPClot™ prototype tubes at RT and 50 °C (Figure 5C). The S2238 assays showed that the wet RAPClot™ prototype tubes (Wet 4 °C) retained over 86% of the enzymatic activity at 4 °C for 251 days, significantly lower than 93% of the enzymatic activity in the RAPClot™ prototype tubes (Dry-RT) at RT (Figure 5C). The 50 °C storage for the RAP prototype tubes (Dry-50 °C) caused the loss of approximately 22% of the enzymatic activity at day 251 compared with the RAP-RT tubes, similar to that of the E-beam-sterilized RAPClot™ prototype tubes (Dry-EB RT) (Figure 5C). The E-beam sterilization plus 50 °C storage caused the RAPClot™ prototype tubes (Dry-EB-50 °C) to decrease activity by ~50% over this time period (Figure 5C). The blood clotting activities of the RAPClot™ tubes with different treatments under different storage conditions paralleled those of the S2238 assay (Figure 5C). The E-beam-sterilized RAPClot™ prototype tubes at 50 °C storage (Dry-EB-50 °C) clotted 6 mL of recalcified citrated whole blood only at 109 s and the heparinized blood at 8 U/mL faster than 3 min (174.5 s) (Figure 5C). Furthermore, we prepared four sets of RAPClot™ prototype tubes (A1, A2, B1, and B2) containing high dose (0.9 mU/tube) and one set of low dose RAPClot™ prototype tubes (0.4 mU/tube) as a positive control in protective formulations (Figure 5D). All the RAPClot™ tubes after the E-beam sterilization (25 kGy) were stored at RT for 682 days and then used to clot recalcified citrated whole blood with or without heparin (8 U/mL) (Figure 5D). All the A1, A2, B1, and B2 tubes showed high blood clotting activities, clotting the non-heparinized blood within 2–3 min and the heparinized blood (8 U/mL) in ~4 min (Figure 5D). The positive control RAPClot™ tubes (0.4 mU/tube) clotted the non-heparinized blood at 4 min and the heparinized blood (8 U/mL) slightly over 5 min with high-quality serum produced (339.5) (Figure 5D). The data suggest that the E-beam sterilization retains 30% higher clotting activity in the RAPClot™ prototype tubes compared to the gamma-radiation which showed a clotting activity of ~5 min after storage at RT for nearly two years.

### 3.6. Initial Clinical Study of RAPClot™ Rapid Serum Prototype Tubes

In a small clinical study, the activity of the γ-sterilized RAPClot™ prototype tubes (0.4 mU/4 mL blood) in clotting both fresh and recalcified citrated whole blood from five volunteers was determined (Figure 6). The average clotting times for fresh whole blood for the SST tubes were 455.0 ± 42 s, for the RAP-Ir tubes only 142.5 ± 42 s, for the RAP+Ir tubes 188.5 ± 31.6 s, and for the RAP+Ir+HEP (8 U heparin/mL) 139 ± 36.3 s (Figure 6A). The average clotting times of recalcified citrated whole blood for the RAP+Ir tubes was 95.2 ± 5 s and for the RAP+Ir+HEP was 107.6 ± 10 s (Figure 6B), providing further support that RAPClot™ with γ-radiation clotted efficiently not only recalcified citrated whole blood, but also heparinized blood at 8 U/mL. Furthermore, TEG demonstrated a similar clotting pattern to that of visual clotting in the RAP+Ir tubes with or without heparin (Figure 6C,D). The TEG images are provided here for two volunteers who are representative of all five individuals (Figure 6C,D). RAP+Ir and RAP+Ir+HEP produced R and K times that were 45 and 50 s for volunteer 1 and 50 and 50 s for volunteer 2 while Ca^+^ alone had 870 and 480 s for volunteer 1 blood and 590 and 215 s for volunteer 2 (Figure 6C,D). The study showed that significantly greater angle-α, which measures the speed of clot formation, was obtained in RAP+Ir (79.6 and 77.5) and RAP+Ir+HEP (68.7 and 67.2) compared with Ca^+^ alone (25.4 and 47.3) (Figure 6C,D). MA values in RAP+Ir (75.9 and 59.2), which determines clot strength, were substantially higher, compared with those in Ca^+^ alone (40.0 and 52.6) and RAP+Ir+HEP (35.7 and 35.5) (Figure 6C,D). Figure 6E shows that the average R time of the five volunteers was 641.5 s in Ca^+^, which was significantly decreased to 54.4 s in RAP+Ir and 95 s in RAP+Ir+HEP. Similarly, the K time was 262.4 s in Ca^+^, which was significantly decreased to 63 s in RAP+Ir and 113.2 s in RAP+Ir+HEP (Figure 6F). Angle-α was significantly increased in RAP+Ir (75.3) and RAP+Ir+HEP (64.5) compared with Ca^+^ (43.6) (Figure 6G). The MA value was increased in RAP+Ir (59.3), compared with those in Ca^+^ (47.2) and in RAP+Ir+HEP (38.2) (Figure 6H), suggesting further that RAPClot™ improved substantially the clot strength in normal blood, with a decrease in clot strength in the presence of heparin. Figure 6I shows the V-curves from two participants. V-curves generated by RAP+Ir are similar to those by RAP+Ir+HEP, but significantly different from that by Ca^+^ alone (Figure 6I). The average MRTG of the five participants was 22.5 mm/min in RAP+Ir, significantly higher than those both in RAP+Ir+HEP (13.3 mm/min) and Ca^+^ alone (5.6 mm/min) (Figure 6J). The MRTG in RAP+Ir+HEP was also significantly higher in Ca^+^ (Figure 6J). On average, RAP+Ir had a TMRTG (1.9 min) significantly shorter than RAP+Ir+HEP (3.3 min) and Ca^+^ (13.7 min) (Figure 6K). Total thrombus generation (TTG) in RAP+Ir was significantly higher than that in RAP+Ir+HEP, but no significant difference was statistically obtained between RAP+Ir and Ca^+^ and between RAP+Ir+HEP and Ca^+^ (Figure 6L).

### 3.7. Analyte Measurements from Sera Produced in RAPClot™ Rapid Serum Prototype Tubes

We next determined the levels of commonly measured analytes (potassium (K), glucose (Glu), lactate dehydrogenase (LD), iron (Fe), and phosphate (Phos)) known to be affected by clotting and cell lyses during the clotting process, and remnant cells on top of the blood gel “buffy coat” or cells in contact with the serum in gel-free tubes post centrifugation [3]. Blood was collected from the five participants and serum was produced by the three 0.4 mUnit RAPClot™ prototype tubes (RAP-Ir, RAP+Ir, and RAP+Ir+HEP), and compared with that generated by the commercial SST tube (Figure 7). As shown in Figure 7, the values for the five analytes examined were similar in the sera of the five volunteers’ fresh blood produced in four blood collection tubes with *p* values > 0.05. Furthermore, in a separate clinical trial with five volunteers, we compared 33 analyte measurements in the sera generated from the RAP+Ir and RAP+Ir+HEP (8 U/mL) prototype tubes, compared with those from the three commercial blood collection tubes (SST, RST, and PST tubes) (Appendix A). The results showed that there was very good agreement between the analyte concentrations determined among the variation in all the 33 analytes measured among the five tubes (Appendix A). These results suggest that the presence of RAPClot™ in the prototype blood collection tubes does not interfere with analyte determination.

## 4. Discussion

Blood collection tube components starting with the tube wall, and including rubber stoppers, lubricants, anticoagulants, separator gels, clot activators, and surfactants, can all affect the quality of the sample generated and subsequently the accuracy and precision of laboratory tests [32]. Surfactants are usually low-viscosity silicone-based fluids used to coat collection tubes to enhance the spreadability of blood in tubes during blood clotting and act as a release agent to ensure the clean separation of clotted blood from the walls during centrifugation, reducing “clots hang up”. In this study, we used several surfactants to coat both GBO and BD plain blood collection tubes which were used for testing the blood clotting activity of RAPClot™ consistent with published studies. In this study, the surfactant used was 20 µL of DC3771 at the concentrations of 0.25–0.5% used for preparing per RAPClot™ prototype tube in which the findings showed that it did not affect either the S2238 assay or blood clotting activities for the assay of 31 analytes.

Hepes buffer has been widely used for protein purification and extraction as a stabilizer of different proteins [33,34,35,36]. In the present study, Hepes was used for preparing the RAPClot™ aqueous working solution which retained a high level of blood clotting activity. However, when it was used for drying RAPClot™ under different temperature conditions in two types of commercial plain tubes (GBO and BD), the blood clotting activity was significantly reduced whether the tubes were coated with or without surfactant. In contrast, the use of Gelofusine (GF) for preparing and drying the RAPClot™ working solution in both the GBO and BD tubes retained both high enzymatic and blood clotting activities. Therefore, GF was selected for use in the RAPClot™ prototype tubes. It was subsequently found that the inclusion of 10% stabilizing sugar in the GF buffer optimized the recovery and stability of the RAPClot™ tubes and was selected for further studies to optimize the prototype formulation performance.

A requirement for blood collection tubes is that they can be stored for periods in excess of 12 months in order to eliminate stability issues. Gamma-radiation (γ-radiation) is a method of choice and was first used for sterilizing disposable medical products in the 1960s [37,38]. Since then, over 50% of the disposable medical products manufactured in developed countries have been radiation-sterilized at doses varying from 17 to 50 kGy [38,39,40]. Here, we used γ-radiation at the doses of 25–27.5 kGy to sterilize the RAPClot™ prototype tubes. Even at the highest dose, we revealed a recovery of approximately 42% of the enzymatic activity and 53% of the plasma clotting activity in the RAPClot™ tube. While it is evident that the metalloproteinase is sensitive to γ-radiation, the use of the GF formulation limits the loss of activity and retains the usefulness of RAPClot™ as an important additive to blood collection tubes to enhance clotting to produce quality serum. The γ-radiation-sterilized RAPClot™ prototype tubes prepared in all six formations on day 14 post-radiation were capable of clotting heparinized blood at a high dose of 8 U/mL within 5 min even at concentrations as low as 0.26 mU/tube. These data demonstrated that γ-radiation is acceptable for sterilizing RAPClot™ prototype tubes. More recently, E-beam radiation has also been employed to sterilize pharmaceutical packaging products [41]. It operates by directing a continuous flow of electrons through the articles being sterilized [42]. We also used E-beam at 25 kGy to sterilize RAPClot™ prototype tubes, finding that the tubes retained up to 80% of the enzyme activity based on the S2238 assay, revealing that this form of sterilization is an effective technology for developing the RAPClot™ prototype tubes.

The standard shelf life for blood collection tubes is 12 months from the time of manufacture [29,30]. Thus, in introducing any new blood collection tube into the market, this is an important consideration for its commercial viability. Our results showed that RAPClot™ is stable in blood collection tubes prepared using a protective formulation and sterilized with both Y-radiation and E-beam exposure. RAPClot™ stored in blood collection tubes at room temperature in either wet or dried form (evacuated) was stable for periods of up to 3 years. The activity was comparable in both citrated and heparinized (8 U/mL) blood and it was capable of withstanding temperatures of 50 °C for periods as long as 286 days. The capacity to produce quality serum in the presence of anticoagulants is a big advantage over the RST tube and Greiner BCA Fast Clot tube [4] to which thrombin is added and have been found not to be suitable for serum preparation for patients on anticoagulants, especially heparin [3]. The activity of the RAPClot™ tube was also maintained when exposed to sterilization conditions at levels used commercially. While the more conventional method of exposure to γ-radiation sterilization reduced activity by 50%, exposure to E-beam radiation only had minimal effects on its activity.

The results presented here demonstrate that RAPClot™ can be used to produce RAPClot™ rapid serum prototype tubes that can meet the requirement of a new standard blood collection tube to clot blood samples including high-dose heparinized blood (8 U/mL) within 5 min post-γ-radiation or E-beam sterilization with a potential for long-term storage stability at RT. Further clinical testing of spray-dried and irradiated RAPClot™ rapid serum tubes is warranted.

While the RAPClot™ tube possessed the major characteristics required for a suitable serum tube, it was also important that the presence of RAPClot™ did not lead to any interference with analytes tested in a clinical setting. Accordingly, we carried out two clinical trials with five volunteers to test whether there was any interference with biochemical analytes in these tubes. In the initial trial, the concentrations of five commonly measured analytes: K, Glu, LD, Fe, and Phos by the three RAPClot™ tubes (RAP-Ir, RAP+Ir, and RAP+Ir+HEP) were equal to those generated by the commercial SST tube. In a separate clinical trial with the five participants, the concentrations of 33 biochemistry markers in sera were generated from the two RAPClot™ tubes, which were not clinically different to those generated from the commercial SST, RST, and PST tubes. In fact, the K and Gluc in the serum from heparinized blood were closer to lithium heparin plasma. These data provide strong evidence that the RAPClot™ prototype tube is suitable for routine laboratory use in both non- and heparin-anticoagulated blood samples.

## 5. Conclusions

In conclusion, we have developed a novel blood collection tube (RAPClot™ rapid serum prototype tube) that produces quality serum for analyte determination. We established a stabilizing formulation that upon spray-drying as an additive in commercial plain tubes and radiation using standard commercial methods and levels retained RAPClot™ enzyme and blood clotting activities when stored over periods of time required for commercial viability including samples with very high concentrations of heparin. The experimental data have demonstrated that a RAPClot™ containing clotting tube is a commercially competitive product and has the potential to be the standard tube, eliminating the need for the various serum/plasma tubes, e.g., SST, RST, BCA Fast, PST (gel or plasma separator device), Na fluoride, and FC Mix due to its ability to clot all anticoagulated samples within five minutes and consistently produce the highest quality serum and significantly minimize sample quality related result errors with optimal TAT.

## Figures and Tables

**Figure 1 biomolecules-14-00645-f001:**
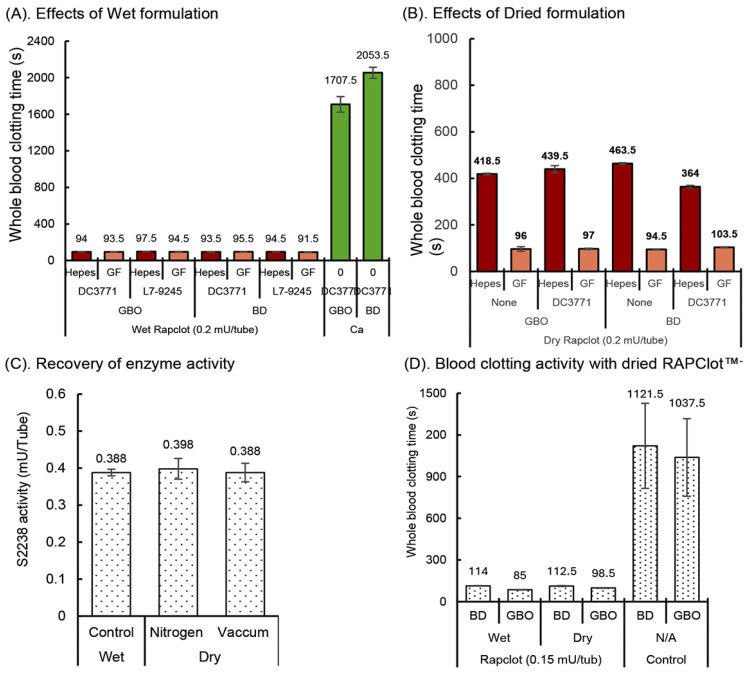
Effects of exogenous conditions on the development of RAPClot™ -prototype blood collection tube. (**A**). The comparison of RAPClot™ –Hepes working solution with RAPClot™ –Gelofusine working solution in the presence of the surfactants DC3771 or L7-9245 when clotting 4 mL recalcified citrated whole blood. (**B**). The activity of air-dried RAPClot™ prepared from Hepes and Gelofusine solutions with or without the surfactant DC3771 in clotting 4 mL recalcified citrated whole blood. (**C**). The S2238 activity of wet RAPClot™ (0.4 mU/tube) in GBO plain blood collection tube at 4 °C for 48 h, compared with those of dried RAPClot™ in GBO plain blood collection tube by nitrogen-drying for 2 h, then stored at RT for 46 h and vacuum-desiccator air-dried for 24 h, then stored at RT for 24 h. At 48 h post preparation, the activity of RAPClot™ in the three groups of the RAPClot™ -containing tubes was analyzed in a single reaction mixture containing prothrombin and the thrombin-specific substrate S2238 (*n* = 6). (**D**). The activity of RAPClot™ (0.15 mU/tube) in stabilizing formulation in both the BD and GBO plain tubes vacuum-desiccator air-dried in clotting 4 mL of recalcified citrated whole blood. The error bars represent the standard deviation (SD).

**Figure 2 biomolecules-14-00645-f002:**
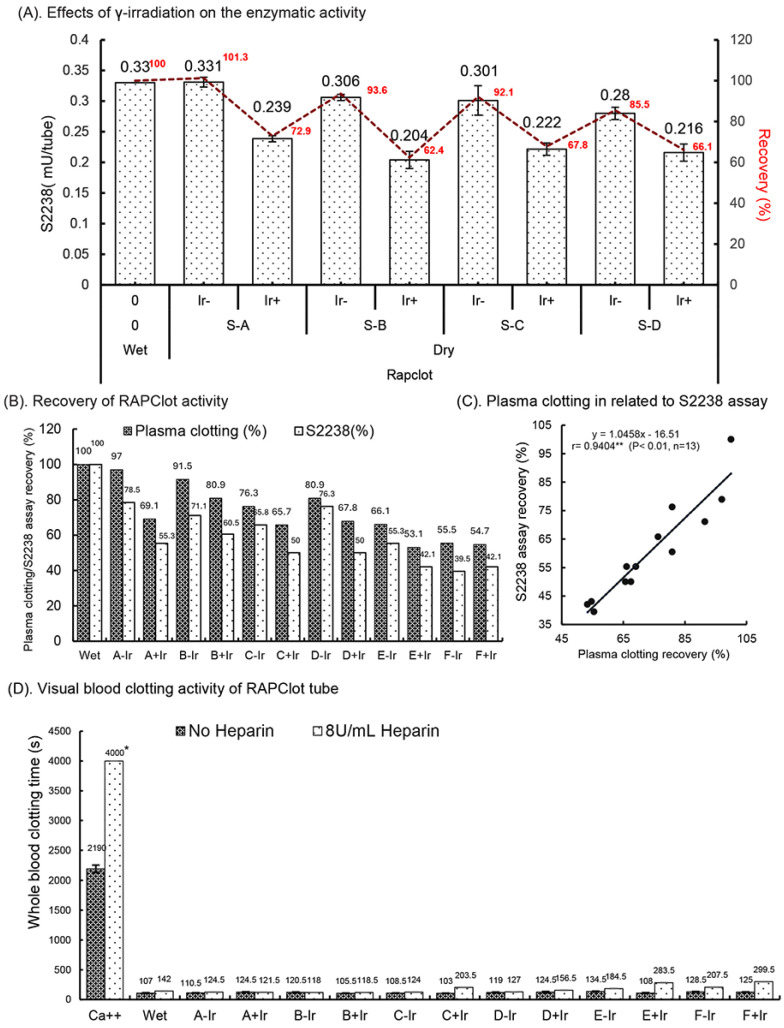
Effects of γ-radiation on the enzymatic activity of the RAPClot™ -containing prototype tubes prepared in different formations in GBO white-top plain tube. (**A**). The recovery (%) of the enzyme activity of the RAPClot™ -prototype tubes (0.33 mU/tube) in four formulations (S-A, S-B, S-C, and S-D) with or without γ-radiation (25.34 kGy) at day 4 post-radiation (*n* = 3) (see also Appendix A). (**B**). The activity recovery (%) of the RAPClot™ -prototype tubes (0.26 mU/tube) in six formulations with or without γ-radiation (27.8 kGy) at day 14 post-radiation in clotting recalcified citrated plasma, compared with the S2238 assay (*n* = 2) (Appendix A). (**C**). The activity recovery (%) of plasma clotting correlated with the S2238 assay for the RAPClot™ -prototype tubes with γ-radiation (27.8 kGy). r = 0.9404 (*p* < 0.01, *n* = 13)) representing a highly significant linear relationship between two assays. ** indicates that *p* < 0.01. (**D**). The activity of the RAPClot™ -prototype tubes prepared in six formulations (**A**–**E**, see Appendix A) with or without γ-radiation (27.8 kGy) at day 14 post-radiation in clotting recalcified citrated whole blood spiked with or without heparin at 8 U/mL. Note: * indicates no clotting occurred (*n* = 2). (**E**). Serum images that were generated from normal and heparinized blood (8 U/mL) clotted in the RAPClot™ prototype tubes with or without sterilization by γ-radiation 24 h after centrifugation (*n* = 2) (Appendix A). The error bars represent the standard deviation (SD).

**Figure 3 biomolecules-14-00645-f003:**
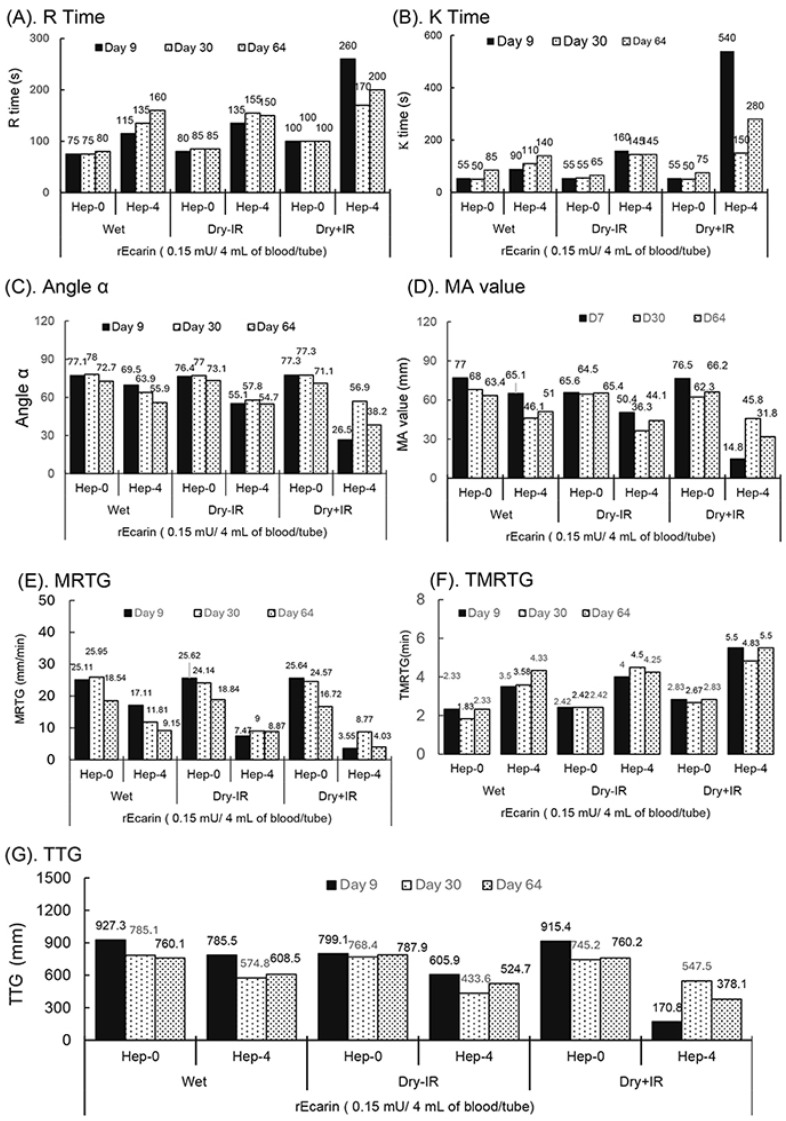
TEG assay showing the stability of RAPClot™ in prototype tube at a dose of 0.15 mU/tube prepared in one formulation with or without γ-radiation (25.7 kGy) stored at room temperature for 64 days in clotting recalcified citrated whole blood with or without heparin (4 U/mL). (**A**). R time, (**B**). K time, (**C**). α angle value, (**D**). MA value, (**E**). MRTG (maximum rate of thrombin generation), (**F**). TMRTG (time to maximum rate of thrombus generation), and (**G**). TTG (total thrombus generation.

**Figure 4 biomolecules-14-00645-f004:**
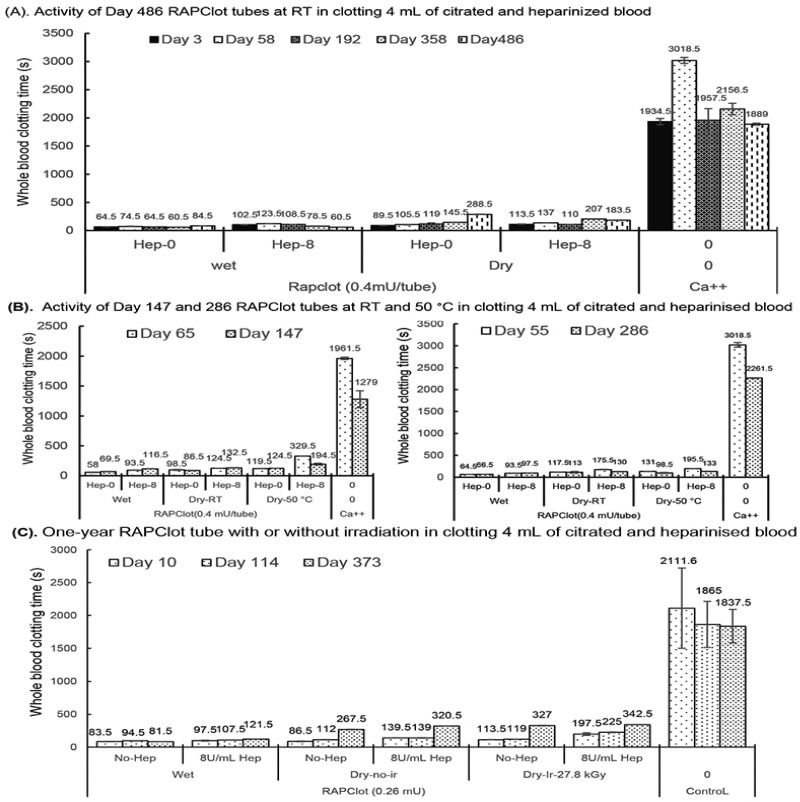
Blood clotting activity of the RAPClot™ prototype tubes with or without γ-radiation that were stored at room temperature (RT) or 50 °C in the long term. (**A**). The activity of the RAPClot™ prototype tubes stored at RT for 486 days in clotting 4 mL recalcified citrated whole blood with or without heparin (8 U/mL). (**B**). The activity of the RAPClot™ prototype tubes (0.4 mU/tube) stored at RT or 50 °C for 147 days (left panel) or 286 days (right panel) in clotting 4 mL recalcified citrated whole blood with or without heparin (8 U/mL). (**C**). The activity of the RAPClot™ prototype tubes (0.26 mU/tube) with or without γ-radiation (27.8 kGy) that were stored at RT for 373 days in clotting 4 mL recalcified citrated whole blood with or without 8 U/mL heparin. The error bars represent the standard deviation (SD).

**Figure 5 biomolecules-14-00645-f005:**
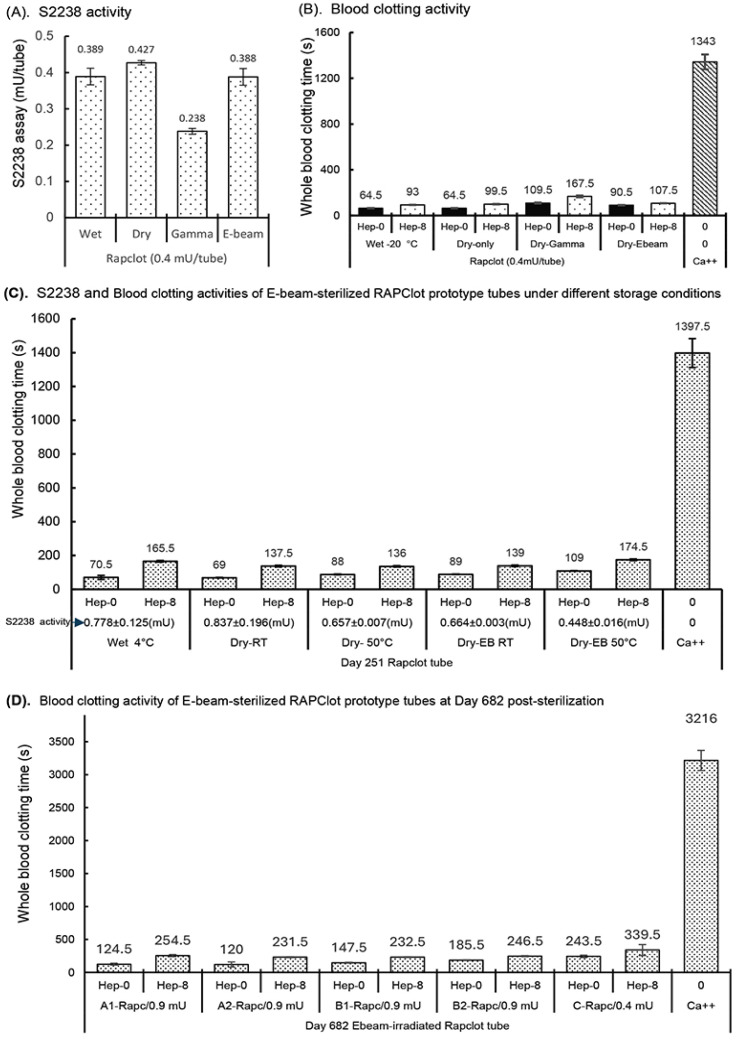
Enzymatic and blood clotting activities of the RAPClot™ prototype tubes with E-beam radiation. (**A**) The S2238 assay showed the recovery of enzyme activity of the RAPClot™ prototype tubes with the E-beam treatment compared with those with γ-radiation at day 7 post-radiation (*n* = 4). (**B**). The activity of the RAPClot™ prototype tubes with the E-beam treatment in clotting recalcified citrated whole blood (4 mL of blood/tube), compared with those with γ-radiation at day 7 post treatment (*n* = 2). (**C**). The enzymatic and blood clotting activities of the RAPClot™ prototype tubes (0.9 mU of RAPClot™ was added to one tube at day 0. Its enzymatic and blood clotting activities were measured at day 251 post E-beam sterilization) in clotting recalcified citrated whole blood (6 mL of blood/tube) with or without heparin (8 U/mL) (*n* = 3). Blue arrow indicates enzymatic activity of the individual RAPClot™ tubes based on S2238 assay at Day 251 post-E-beam sterilization. (**D**). The blood clotting activities of the RAPClot™ prototype tubes at day 682 post E-beam sterilization in clotting recalcified citrated whole blood (4 mL of blood/tube) with or without heparin (8 U/mL) (*n* = 2). Tube A1, A2, B1, and B2 contained 0.9 mU of the RAPClot™ prototype tube at day 0. As a positive control, the amount of RAPClot™ at 0.4 mU/tube was reduced from 0.9 mU/tube to achieve optimal conditions. The error bars represent the standard deviation (SD).

**Figure 6 biomolecules-14-00645-f006:**
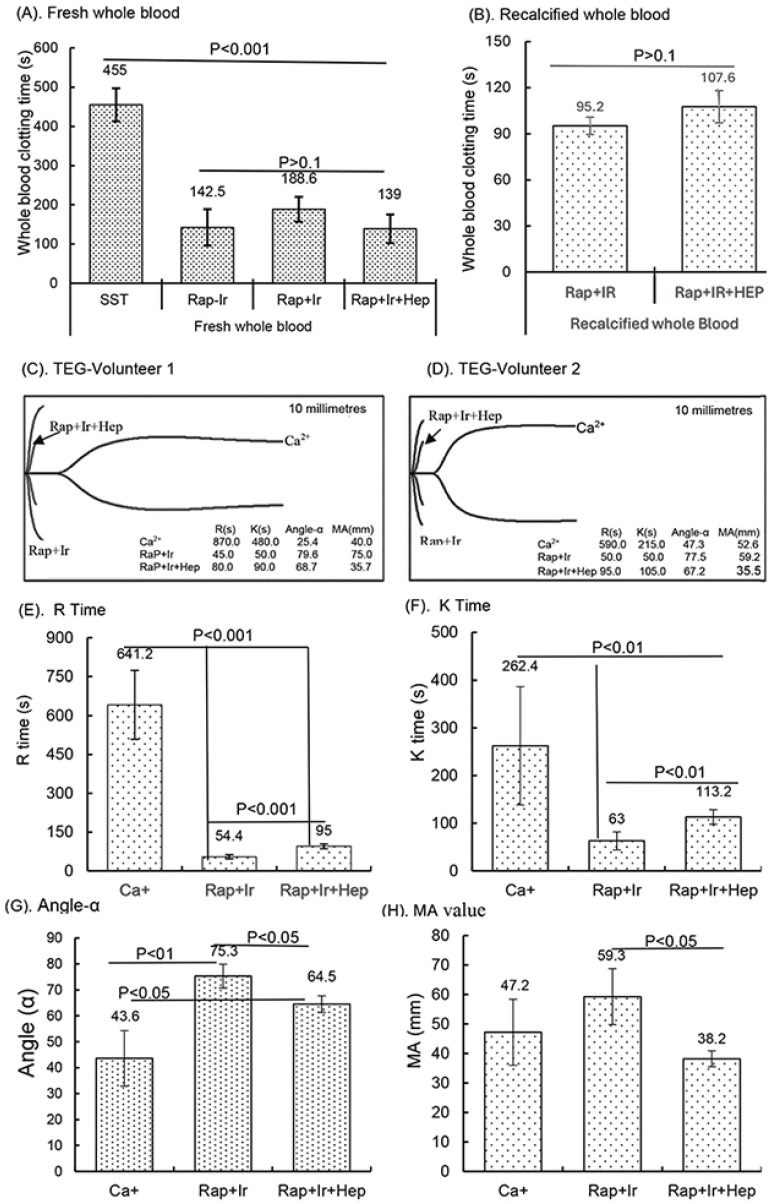
Clinical blood clotting trial with the RAPClot™ tubes using the five participants. The activity of the RAPClot™ tube in clotting fresh whole blood and recalcified citrated whole blood samples from the five volunteers by visual clotting and TEG assays. A total of 0.26 mU of RAPClot™ was used to add into one GBO plain tube for preparing the RAPClot™ prototype tube with or without γ-radiation as described in the Material and Methods Section. (**A**). The visual clotting activity of the BD SST tube (SST), RAPClot™ prototype tube without radiation treatment (Rap-Ir), and RAPClot™ prototype tube treated with r-radiation (Rap+Ir) in clotting fresh whole blood and the RAPClot™ prototype tube treated with γ-radiation (Rap+Ir+Hep) in clotting fresh whole blood containing heparin at 8 U/mL. (**B**). The activity of RAPClot™ prototype tubes with r-radiation treatment in clotting recalcified citrated whole blood without (Rap+Ir) or with heparin at 8 U/mL (Rap+Ir+Hep). The data are the mean ± standard deviation (X ± SD) from the 5 volunteers. (**C**,**D**). Parallel to the visual clotting assay shown in (**B**), the thromboelastography (TEG) assay shows two representative images of RAPClot™ in clotting recalcified citrated whole blood samples from the three types of tubes (Ca^+^, Rap+Ir, and Rap+Ir+Hep) from 2 volunteers out of the 5 volunteers, revealing significantly different thromboelastographic traces of the RAPClot™ prototype tubes from Ca^+^-only tubes in clotting recalcified citrated whole blood and different TEG parameters (R times, K time, α angle values, and MA values) showing in inserted squares. (**E**–**H**). TEG assay shows four parameters (R times, K time, α angle values, and MA values) of the RAPClot™ prototype tubes in clotting recalcified citrated whole blood from five participants. The data are the x ± SD of the four parameters from the five participants in three types of tubes in the duplicate assay (*n* = 10). *p* < 0.05, *p* < 0.01, and *p* < 0.001 represent that the difference between the two types of tubes was statistically significantly different, respectively. *p* > 0.1 represents that the difference between the two types of tubes was not statistically different. (**I**). Two representative V-curves derived from the TEG assay for two volunteers (see Figure 1C,D) showing significantly different thrombin generation of the RAPClot™ prototype tubes in clotting recalcified citrated whole blood with or without heparin. (**J**). The mean ± SD of the maximum rate of thrombin generation (MRTG) at mm/min of the recalcified citrated whole blood from the 5 participants clotted in three types of tubes in the TEG assay in duplicate (*n* = 10). (**K**). The mean ± SD of the time to the maximum rate of thrombus generation (TMRTG) (min) (*n* = 10). (**L**). The mean ± SD of the total thrombus generation (TTG) (*n* = 10). *p* < 0.05, and *p* < 0.01 represent that the difference between the two types of tubes was significantly different, respectively. The error bars represent the standard deviation (SD).

**Figure 7 biomolecules-14-00645-f007:**
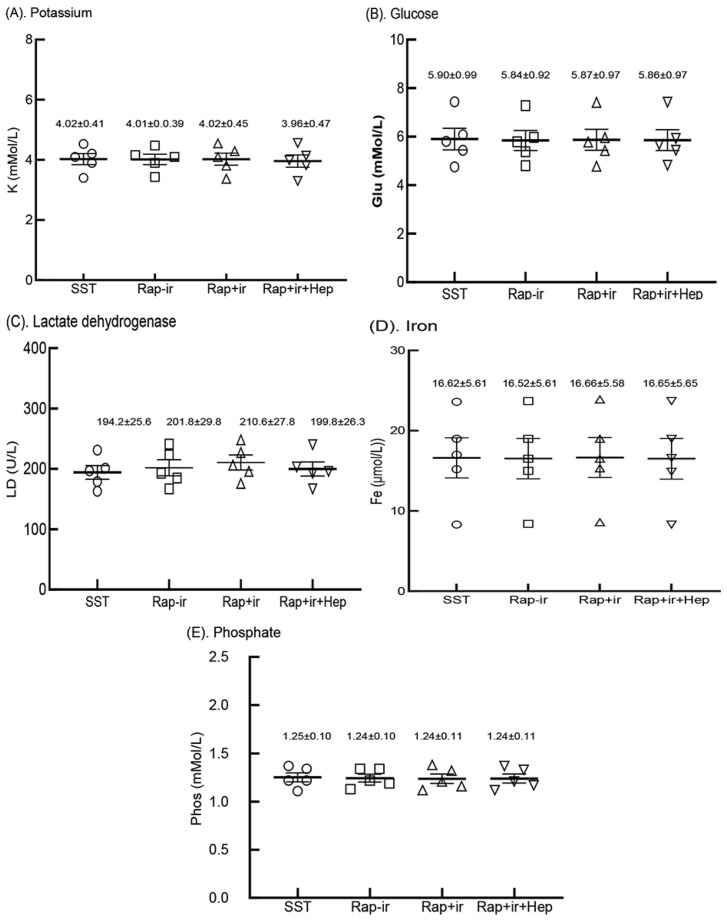
Five analytes in the sera of the five volunteers’ fresh blood clotted in the BD SST tube (SST), RAPClot™ prototype tube without γ-radiation (Rap-Ir), and RAPClot™ prototype tube treated with γ-radiation (Rap+Ir) in clotting fresh whole blood and RAPClot™ prototype tube treated with γ-radiation (Rap+Ir+Hep) in clotting heparinized fresh whole blood (8 U/mL) in the clinical trial as shown in Figure 6A. (**A**). Potassium (K) (*p* = 0.9999, >0.05), (**B**). glucose (Glu) (*p* = 0.9997, >0.05), (**C**). lactate dehydrogenase (LD) (*p* = 0.8191, >0.05), (**D**). iron (Fe) (*p* = 0.9997, >0.05), and (**E**). phosphate (Phos) (*p* = 0.9966, >0.05). The data inserted in the individual sub-figures are the mean ± standard deviation (SD) of the five volunteers, respectively (*n* = 5). The data analysis results did not show that the concentrations of the individual analytes among the four tubes were statistically different (*p* > 0.05). The error bars represent the standard deviation (SD).

## Data Availability

We confirm that the data supporting the findings of this study are available within the article and its Appendix A.

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
