# Peer review of "Generation of Rapid and High-Quality Serum by Recombinant Prothrombin Activator Ecarin (RAPClot™)"

_biomolecules, 2024, doi:10.3390/biom14060645_

Round 1

Reviewer 1 Report

Comments and Suggestions for Authors

per attached file

Comments on the Quality of English Language

generally well written

Author Response

Response to Reviewer 1 Comments

1.    Summary

3.      Point-by-point response to Comments  

Thank you for pointing this out. We agree with this comment. Therefore, we have revised our manuscript based on your comments and suggestions.

1). Science

a). The authors repeatedly speak about high quality (or highest quality) serum. However, no indication is made about how the quality is determined, measured or monitored.

Response 1. Quality serum is generated post complete clotting of blood and in the process it is depleted of fibrinogen, fibrin strands, cells or cell stroma. The most sensitive markers are potassium, LDH and Glucose. In cells potassium is about 25 higher than plasma concentration. During the clotting process cells are raptured hence produce potassium concentrations with higher levels than lithium heparin plasma. Similarly, LDH is present in cells and the rapture of cells releases LDH, falsely increasing the serum concentration. Presents of cells in the serum upon storage will consume glucose, decreasing the concentration over time. The presence of fibrin or cell stroma can interfere with assays in a several ways. The fibrin may bind analytes, change the test sample volume or cause spectral interference. Likewise effects may be encountered with cellular stroma.This has been included in our results section page 5, line 27-35 (Highlighted in yellow) and has involved the inclusion of a new supplementary figure (Supp Fig. 3 and 4), showing that the quality of RAPClot-produced serum is considerably better than that produced in commercial tubes.  Furthermore, we have demonstrated in Supplementary table 3 that all 33 not just potassium, LDH and glucose, biochemical analytes determined in RAPClot-produced serum are consistently similar to those produced in commercial tubes. 

  1. The authors mention that serum prepared, in currently available tubes for this purpose, frequently contain some cellular fragments. However, no effort is made to measure or quantify these.

Response 2. As we referred to above, we have included Supplementary figure 3 and 4, which visually demonstrates that serum produced in RAPClot tubes is cleaner (blood is completely clotted, insignificant levels of fibrinogen or fibrin and cell or cell stroma present. On the other hand, in Supplementary figure 3 and 4, it is clear that serum produced in a commercial tube contains red blood cells, other cellular material and fibrin strands. It is also evident from Supplementary figure 3 and 4 that the serum produced in several commercial blood collection tubes is of low quality compared to that in a RAPClot tube. 

  1. Particulate matter can be measured with instruments such as DLS (dynamic light

scattering) based particle sizers, which can characterize both the relative concentration and particle sizes.

Response 3. As we have pointed above, our visual assessment demonstrates clearly difference in quality of serum produced in a RAPClot tube compared to commercial tubes. We feel that dynamic light scattering is unnecessary here, particularly since there is no interference with biochemical determinations, as revealed in assays for 33 biochemical analytes (Supplementary table 3).

iii. Cellular debris should be differentiated from microparticles or exosomes. The latter may well be present in serum under certain conditions. It would be of value to find out if these are still present in the serum prepared by the RAPClot™ method. But this would require some objective particle analytic measurements.

Response 4. Again, as we have referred to above, when 33 analytes were determined in RAPclot serum, there was no interference. This strongly suggests that the serum does not contain particulate matter that would cause interference.  

  1. b) 7. Figure 2A: The error bars are presented for all but the first measurement. The figure

legend does not indicate the value of n. Similarly, there is no indication in the legend what the different formulations are. Presumably, the Supplementary Table 1 is involved. But this should be stated.

Response 5. Error bars were not included for the first measurement since all three values were the same. The number for n=3 has been added to the legend. Reference to different formulations is made in Supplementary Table 1 which was already referred to in the text (Top of Page 7. Changes are highlighted in yellow).

  1. c) 7. Figure 2B: The RAPClot activity recovery under 6 different formulations is presented. However, no indication of either n or formulation that is represented. Presumably the Supplemental Table 2 is involved. However, this is not stated clearly in the legend.

Response 6. We have made the additions to the legend which are highlighted in yellow (Top of Page 9).  The different formulations are already shown at the bottom of the figure. Figure 2A has been changed to Figure 2B in the text as highlighted in yellow.

  1. d) 7. Figure 2D: Whole blood clotting times are obtained with 6 different formulations. The legend should refer the formulations to Supplementary Table 2, instead of Supplementary Table 1.

Response 7. Supplementary Table 1 has been changed to Supplementary Table 2 as highlighted in yellow in figure 2 legend.

2) Presentation: This manuscript is generally well written. The following are examples.

that can be improved.

  1. a) Acronyms:

i.1.31: Change “LD” to “lactate dehydrogenase (LD)”.

Response 8. Taking the reviewer’s suggestion, we have changed “LD” to “lactate dehydrogenase (LD) (Abstract, Page 1, highlighted in yellow)

  1. 2.48: Change “AT111” to “antithrombin III (ATIII)”.

Agreeing with the reviewer’s suggestion, we have changed “AT111” to “antithrombin III (ATIII)” (Paragraph 4, Page 2, highlighted in yellow)

  1. b) 5: Figure 1

  1. legend.2-3: Change “working solution in clotting 4mL recalcified” to “working solution,

in the presence of surfactants DC3771 or L7-9245, when clotting 4mL recalcified.”

Response 9. We have changed to that working solution in the presence of surfactants DC3771 or L7-9245, when clotting 4mL recalcified (Page 6, Fig 1A legend, highlighted in yellow).

  1. legend.4: Change “from Hepes and Gelofusine solutions in clotting 4mL recalcified”

to “from Hepes or Gelofusine solutions, with or without surfactant DC3771, in clotting

4mL recalcified”

Response 10. We have revised the legend according to the reviewer’s suggestion (Page 6, Fig 1B legend, highlighted in yellow).

  1. c) 8: Figure 2
  2. Legend.12-13: Change “clotted in none or γ-radiation sterilized RAPClot prototype.

tubes 24 hr after” to “clotted in RAPClot prototype tubes, with or without sterilization.

by γ-radiation, 24 hr after

Response 11. We have revised the legend according to the reviewer’s suggestion (Page 9, Fig 2E legend, highlighted in yellow).

Reviewer 2 Report

Comments and Suggestions for Authors

Generation of Rapid and High-Quality Serum by Recombinant review 

No 

Present 

Comment 

1

Key word, 7 

To be minimize 

Introduction 

Font size of references 

It is different to be uniform it 

Material and method

There is no study design, time of study, duration, place 

Material and method

There is no statistical method?? 

Material and method

Which p value used and at which level 

Results 

What is type of figure, is it histogram? 

Results 

Figure resolution is not clear 

Results 

Many words in figures need to be re writing 

Results 

What about sensitivity and specificity of the results as compare to other method which used different tube 

Conclusion 

It needs to be summarize 

Conflict of interest 

The authors are employer in the  Q sera company 

References 

Many references are from 2000 and older ? 

Comments on the Quality of English Language

Dear Author 

you need to uniform font size, use more clear resolution figures and use better word to reflect your ideas 

Author Response

Response to Reviewer 2 Comments

1.    Summary

2.     Questions for General Evaluation

Reviewer’s Evaluation

Does the introduction provide sufficient background and include all relevant references?

Yes

Are all the cited references relevant to the research?

Yes /Can be improved

Is the research design appropriate?

Yes

Are the methods adequately described?

Yes/Can be improved

Are the results clearly presented?

Yes

Are the conclusions supported by the results?

Yes

  1. Point-by-point response to Comments  

Key word

.  

Response 1 Seven key word are acceptable

Materials and Method

  1. There is no study design, time of study, duration, place.

Response 2. We have incorporated the following into Materials and Methods section to a new paragraph of study design (Page 3 and 4, Highlighted in yellow).

To use RAPClot as an additive to develop a novel type of blood collection tube for clinical diagnosis, we designed and carried out different types of experiments in the laboratory. A). Formulation development experiments: different colloids such as lactulose, dextran, Polyvinylpyrrolidone, voluven and sorbitol were initially tested for developing an optimal RAPClot formulation which is substantially required for clotting anticoagulant-blood, especially heparin-blood to produce high quality serum (Supplementary Figure 2).  At last, six RAPClot formulations were designed for developing RAPClot prototype tubes for further experiments (Supplementary Table 1 and 2). B). γ-radiation and Electron-beam (E-beam) sterilization experiments: RAPClot prototype tubes prepared in different formulations were dried in nitrogen air at room temperature.  The RAPClot prototype tubes were then divided into two groups, one group of the tubes was sterilised by either γ-radiation (Supplementary Table 1 and 2) or E-beam (Figure 5) with typical dose range of 25-30 kGy, and the other group was used as control without γ-radiation and E-beam sterilization. Both types of the prototype tubes were assayed for activity of RAPClot by S2238 assay and blood clotting assay. C). Shelf-life experiments of RAPClot prototype tubes: Currently, most blood collection tubes on the market have at least 12-month shelf-life (Ref). Thus, the prototype tubes whether they were dry-only or sterilized by γ-radiation/E-beam were also divided into two groups, which were respectively stored under two temperature conditions- room temperature (RT) and higher temperatures (50 °C) that could cause reductions in draw volume up to two years. D). Small clinical trials: Trial 1 was designed to recruit five volunteers for assessing capacity of γ-sterilized RAPClot™ prototype tubes in clotting both fresh and heparinised blood and its effects on determination of five important analytes. Trial 2 was designed to determine all 33 analytes in serum sera produced by RAPClot tubes from the five volunteers, compared to those produced by three commercial blood collection tubes. 

  1. There is no statistical method.

Response 3. Agreeing with the reviewer’s comment, we have added a section on “Statistical methods” in Materials and Methods section (A new paragraph is highlighted in yellow, Page 4).

  1. Which p value used and at which level.

Response 4. P values at P<0.05 and P< 0.01 were used to present the levels of significance in Statistical methods (A new paragraph is highlighted in yellow, Page 4).

Results

  1. What type of figure, is it histogram?

Response 5. Yes, most of the figures are histograms.

  1. Figure resolution is not clear.

Response 6. The figure quality is in line with requirements of the journal but we are happy to provide high resolution figures.

  1. Many words in figures need to be rewritten.

Response 7. We have made revisions for the figure legends.

  1. What about sensitivity and specificity as compared to other methods which used different tube

Response 8. As we reported previously(Zhao et al, Biomolecules 2022, 12, 1704), RAPClot tube is substantially higher sensitivity in clotting not only normal whole blood but also the blood containing all anticoagulants such as  heparin, warfarin, dabigatran, Fondaparinux, rivaroxaban and apixaban to produce clean serum for accurate analyte determination in diagnostic medicine compared to other blood collection tubes containing specific additives such as the EDTA tube, lithium heparin tube and Thrombin-containing tubes (RST) that are used for specific analysis. For example, the EDTA-tube is used for most hematology procedures such as identifying or counting blood cells, blood type etc. The lithium heparin tube is used to produce plasma for clinical chemistry tests such as cholesterol, CRP, hormones etc.  RST tube has been developed to speed up the blood clotting process when producing high quality serum for blood testing in a routine diagnostic laboratory setting, but not for heparinized blood (Dimeski, et al, Clin. Chem. Lab. Med. 2010, 48, 651–657).

Conclusion

1.Conflict of interest: The authors are employer in the Q-sera company?

Response 9. As indicated under affiliations on page 1 of the manuscript only Michael Grant is an employee of Q-Sera. All of the other authors are employees of The University of Queensland or the Princess Alexandra Hospital.

  1. Reference: Many references are from 2000 and older?

Response 10. We disagree with the reviewer, 36 of the 42 references cited are 2000 or more recent and of these 26 are 2010 or later. Only 6 are prior to 2000 and these refer to original methodology or background data.

Round 2

Reviewer 1 Report

Comments and Suggestions for Authors

per attached file

Author Response

Reviewer 1

2) Presentation: This manuscript is written very well. Minor corrections remaining.

  1. a) 3.33-34: Change “which is substantially required for clotting anticoagulant-blood,

especially heparin-blood to produce” to “which is particularly helpful for clotting.

anticoagulated blood, especially heparinized blood, to produce.”

Response 1. We have revised the sentence according to the reviewer’s suggestion (Page 3, line 5-6, Study design section, highlighted in blue).

  1. b) 3.35: Change “At last, six RAPClot formulations” to “Finally, six RAPClot formulations”

Response 2. We have revised “At last” to be “finally” according to the reviewer’s suggestion (Page 3, line 6, Paragraph 1, highlighted in blue).

  1. c) 4.6: Change “analytes in serum sera produced” to “analytes in sera produced”

Response 3. We have deleted the word “serum” from the sentence according to the reviewer’s suggestion (Page 4, line 7, Study design section, highlighted in blue)

  1. d) Figures: It is customary for error bars to represent standard error, unless otherwise

annotated. With a few exceptions, figure legends do not describe what the error bars

represent. However, in the text on page 4, under ‘Statistical analysis’ the authors state that these represent standard deviations, at least for Figures 5 and 7. It would help, if the figure legends stated what the error bars represent

Response 4. Taking the reviewers’ suggestion, we have consistently stated that the error bars represent standard deviations (SDs) in legends of six figures (Figure 1, 2, 4, 5, 6, &7, highlighted in blue)

Reviewer 2

No new comments

Reviewer 2 Report

Comments and Suggestions for Authors

thank you for your effort 

Author Response

(The authors gave the same response as above.)
